# SARS-CoV-2 Viral Load Analysis at Low and High Altitude: A Case Study from Ecuador

**DOI:** 10.3390/ijerph19137945

**Published:** 2022-06-28

**Authors:** Esteban Ortiz-Prado, Katherine Simbaña-Rivera, Raul Fernandez-Naranjo, Jorge Eduardo Vásconez, Aquiles R. Henriquez-Trujillo, Alexander Paolo Vallejo-Janeta, Ismar A. Rivera-Olivero, Tannya Lozada, Gines Viscor, Miguel Angel Garcia-Bereguiain

**Affiliations:** 1One Health Research Group, Faculty of Medicine, Universidad de las Américas, Quito 170124, Ecuador; ksimbanarivera@gmail.com (K.S.-R.); raul.fernandez.n@gmail.com (R.F.-N.); jevasconez@udlanet.ec (J.E.V.); henriquezmd@gmail.com (A.R.H.-T.); ismar.rivera@udla.edu.ec (I.A.R.-O.); 2Department of Cell Biology, Physiology and Immunology, Faculty of Biology, Universitat de Barcelona, 08028 Barcelona, Spain; gviscor@ub.edu; 3Laboratorio de Investigación, Universidad de las Américas, Quito 170124, Ecuador; pvallejoj@gmail.com (A.P.V.-J.); tannya.lozada@udla.edu.ec (T.L.)

**Keywords:** high altitude, COVID-19, SARS-CoV-2, hypoxia, viral load, cycle threshold (Ct), RT-PCR

## Abstract

SARS-CoV-2 has spread throughout the world, including remote areas such as those located at high altitudes. There is a debate about the role of hypobaric hypoxia on viral transmission and COVID-19 incidence. A descriptive cross-sectional analysis of SARS-CoV-2 infection and viral load among patients living at low (230 m) and high altitude (3800 m) in Ecuador was completed. Within these two communities, the total number of infected people at the time of the study was 108 cases (40.3%). The COVID-19 incidence proportion at low altitude was 64% while at high altitude was 30.3%. The mean viral load from those patients who tested positive was 3,499,184 copies/mL (SD = 23,931,479 copies/mL). At low altitude (Limoncocha), the average viral load was 140,223.8 copies/mL (SD = 990,840.9 copies/mL), while for the high altitude group (Oyacachi), the mean viral load was 6,394,789 copies/mL (SD = 32,493,469 copies/mL). We found no statistically significant differences when both results were compared (*p* = 0.056). We found no significant differences across people living at low or high altitude; however, men and younger populations had higher viral load than women older populations, respectively.

## 1. Introduction

Several months after the COVID-19 pandemic declaration on 11 March 2020, some interrogations still remain unsolved concerning viral transmission and the burden of COVID-19 at high-altitude locations [1,2,3]. The variability and susceptibility of COVID-19 infection throughout the world have been attributed to several factors such as chronic diseases, age, socioeconomic level, and access to medical care within different health care systems, among others [4]; however, the role of different environmental factors such as humidity, temperature, or altitude on the transmission of SARS-CoV-2 are not well-understood [4,5]. Since the onset of the pandemic, some researchers have attempted to elucidate the effects of living at a high altitude on the spread of the SARS-CoV-2 virus or on the severity and the COVID-19-related incidence and mortality rate [6,7,8].

A handful of investigations have suggested that living at high altitude is associated with lower COVID-19 attack and mortality rates; nevertheless, the explanations behind those findings have been poorly understood and still need to be elucidated [4,9,10].

At the planetary level, the pandemic has spread with greater virulence in places with higher population density and worse socioeconomic conditions [11,12,13]. This distribution does not apparently respond to environmental conditions but rather to social determinants. However, it is evident that living in remote places with low population density and harsher weathers could justify, in part, a proposed lower viral transmission and reduced SARS-CoV-2 infection rates at higher altitudes [1,6,7].

The hypotheses surrounding the role of high altitude on SARS-CoV-2 transmission and the impact of the pandemic in these populations can be summarized into three groups. The physiological and biological role of altitude-adapted organisms in relation to virus transmission (i.e., the role of ACE-2 receptors at high altitude), the direct or indirect effects of the environment on virulence (i.e., ozone, UV exposure, or cold on viral transmission), or the epidemiological relationship between sociodemographic factors and COVID-19 incidence (i.e., population density, overcrowding activities, or migration) [1,2,14].

In this sense, the least-studied factor is the role of high-altitude exposure on viral stability or viral transmissibility. A suitable indicator would be the evaluation of viral load using the reverse transcription polymerase chain reaction (RT-PCR) test [15,16].

## 2. Materials and Methods

### 2.1. Study Design

An observational, cross-sectional analysis of two populations located at two different altitudes was completed in Ecuador. The objective of this research project was to diagnose COVID-19 cases in remote areas of the country. Most of these areas had poor medical infrastructure and no access to molecular Rt-qPCR testing. 

### 2.2. Settings

The study was carried out in two Kiwchas indigenous communities from Ecuador: Limoncocha, located at 230 m meters above sea level (low altitude), and Oyacachi, located at 3800 m meters above sea level (high altitude) (Figure 1).

### 2.3. Participants

We included 268 members from both communities to compare exposure at high altitude with those living at low altitude. These communities share the same ancestry background as previously reported, serving as a good genotyped-controlled population [17,18,19]. 

### 2.4. Data Source and Variables

Using an informed consent as well as an epidemiological data recollection sheet form, demographic variables including sex, age, jurisdiction of residence (Limoncocha and Oyacachi communities), as well as symptomology status were obtained. For nasopharyngeal swabs samples, the Center for Disease Control (CDC) 2019-Novel Coronavirus (2019-nCoV) RT-qPCR Diagnostic Panel was used to identify the presence of SARS-CoV-2.

### 2.5. Study and Sample Size

In this SARS-CoV-2-detection program, we used a non-probabilistic convenience sample technique that yielded a total of 77 volunteers from Limoncocha (low altitude) and 191 from Oyacachi (high altitude).

### 2.6. Statistical Methods

Measurements of frequency, central tendency, data dispersion, and absolute differences between groups were calculated for all categorical and continuous variables. An independent *t*-test analysis was used to assess differences between groups. We used a 95% confidence level (α = 0.05).

The hypotheses used can be explained as:
**H** **0.***“Viral load in both jurisdictions are equal”.*
**H** **1.***“Viral load in both jurisdictions are different”.*

### 2.7. RNA Extraction and RT-qPCR for SARS-CoV-2 Detection

Nasopharyngeal swabs were collected on 0.5 mL TE pH 8 buffer for SARS-CoV-2 detection by RT-qPCR following an adapted version of the CDC protocol by using PureLink Viral RNA/DNA Mini Kit (Invitrogen, Waltham, MA, USA) as an alternate RNA extraction method and CFX96 BioRad instrument [15,16,20,21,22,23,24,25]. Briefly, the CDC-designed RT-qPCR FDA EUA 2019-nCoV CDC kit (IDT, Newark, NJ, USA) is based on N1 and N2 probes to detect SARS-CoV-2 and RNase P as an RNA extraction quality control [21,23]. Additionally, negative controls (TE pH 8 buffer) were included as control for carryover contamination, one for each set of RNA extractions, to guarantee that only true positives were reported. For viral loads calculation, the 2019-nCoV N-positive control (IDT, Newark, NJ, USA) was used, providing at 200,000 genome equivalents/μL. This positive control is a plasmid including N1 and N2 viral gene targets sequences, and it is a SARS-CoV-2-positive control recommended by CDC guidelines [16,20,22]. 

### 2.8. Nasopharyngeal Sample Collection

Nasopharyngeal swab samples from each of the patients were performed by trained UDLA personnel. Two physicians were in charge of performing the swabs in both communities. The extraction of the genetic material was performed by well-trained laboratorists from the molecular medicine laboratory of the university [26,27].

### 2.9. Bias

The analysis and interpretation of the data was done by four of the investigators separately to look for discrepancies. Any new findings were reviewed by the entire team, and a unanimous decision was made in the event of differences in the results.

### 2.10. Ethics Statement

Informed written consent was obtained from every patient to use their anonymized results from the study in fulfillment of all legal requirements. 

## 3. Results

### 3.1. Demographic Results

From the total sample, 56.3% (*n* = 151) were men, and 43.7% (*n* = 117) were women. The mean age from the entire cohort was 37.72 years (SD = 14.14). In Limoncocha, the mean age was 35.4 years (SD = 14.32) and in Oyacachi was 38.5 years (SD = 14.20). 

### 3.2. Positive Testing Rates

The total number of positive cases was 108 (40.3%), and *n* = 50 were from Limoncocha and *n* = 58 from Oyacachi (Figure 2). The attack rate at low altitude was 64% while at high altitude was 30.3%.

### 3.3. Viral Load Analysis by Altitude

The mean viral load of the 40.3% (*n* = 108) patients who tested positive was 3,499,184 copies/mL (SD = 23,931,479 copies/mL). For Limoncocha, the average viral load was 144,223.8 copies/mL (SD = 990,840.9 copies/mL) and for Oyacachi, 6,394,789 copies/mL (SD = 32,493,469 copies/mL). We found no statistically significant differences between communities (*p* = 0.056) (Figure 2).

### 3.4. Viral Load Analysis by Age and Sex

Among female patients, no difference in viral load were found (viral load mean = 6,165,595 copies/mL (*p*-value of 0.13)). When comparing low- versus high-altitude viral load values, men from Oyacachi had a higher viral load (mean 2,016,226 copies/mL) than those men living in Limoncocha (mean 241,700 copies/mL), with this difference being statistically significant at a confidence level of α = 0.05 (*p*-value of 0.018) (Figure 3).

### 3.5. Viral Load Analysis by Age

In general terms, there were no statistically significant differences when comparing age groups; however, we found significant differences between those aged 25 to 29 years living at high altitude, whose samples had a higher viral load than those from the low-altitude group (Table 1).

When we grouped the individuals into age groups according to the WHO classification, we found a statistically significant difference between the high-altitude young adult men who had higher viral loads than their low-altitude counterparts (*p*-value of 0.017) (Figure 4).

## 4. Discussion

Our exploratory results found no significant difference between people living at high altitude and those living at low altitude in terms of SARS-CoV-2 viral load. While acknowledging the importance of measuring viral load to understand the hypothesized difference across altitudes, our study involves several uncontrolled variables that undermine our ability to conclude if viral loads are different at higher altitudes. Our report is the first one available in terms of exploring this issue, and according to our results, it seems that altitude might not play an important role affecting viral load among patients living at different altitudes. 

In some investigations, the role that hypobaric hypoxia could have on the viability of the SARS-CoV-2 virus has been analyzed [1,25,28]. In very few reports, it is mentioned that ozone could affect the viability of the virus, humidity could affect transmissibility, and ultraviolet rays (UV lights) could eliminate the virus faster than in other locations where there is less UV light exposure than at high altitude [29,30] (Table 2).

Another factor often linked to the transmission of the virus may be cold weather. Several studies have suggested that the susceptibility of the virus to temperature may be affected by climate, but it is more logical to think that climate influences the behavior of people, the way people socialize, and the places people visit (open or closed spaces), as these factors are more feasible when exploring the association between cold and SARS-CoV-2 transmission [35,36,37].

Although socio-demographic factors are the most relevant in terms of viral transmission, incidence, and COVID-19-related mortality, is plausible that altitude may have an important role in improving the survival of some seriously ill patients who live at high altitudes [38]. Even with the presence of comorbidities, survival among severely ill COVID-19 patients at high altitude seems to be improved when compared to low-altitude patients, probably linked to their adaptation status that might improves oxygenation [38,39].

Most of the reports that have described reduced mortality are observational, population-based studies rather than individual-based [3,6,7,10,40], in other words, on populations but not on individuals. This confers a limitation since important data, such as viral load, have never been measured in high altitudes. A recent report by Arias-Reyes et al., 2021 [9], investigated whether the transmission rate of SARS-CoV-2 differs between low and high altitudes [9]. They found that after using a mathematical SEIR model, the probability of viral transmission is lower at high altitude, concluding that their findings strongly support the hypothesis of decreased SARS-CoV-2 virulence in highlands compared to lowlands [9].

Although the available results suggest lower COVID-19-related mortality at high altitudes, lower viral load cannot be attributed as one causal factor. Viral load differences among low- and high-altitude dwellers is unlikely, mainly since the idiopathic response of each organism towards viral replication depends on immunological and biological factors more than in environmental or socio-demographic differences [41,42,43].

The question of whether the SARS-CoV-2 viral load among high-altitude dwellers is different from that of low-altitude dwellers remains unresolved yet. Although our exploratory results indicates that men at high altitude have higher viral load than men at lower altitudes, the differences in viral load are more a function of age and sex rather than altitude.

## 5. Limitations

The study analyzed viral load from COVID-19 patients at low and high altitude in Ecuador. However, there are multiple factors affecting viral load, including age, timing of the samples, disease severity, and other factors that we were not able to control. Days of sampling relative to infection or symptom onset might have a significant impact on viral load. However, this information was not available. The representativeness and recruitment when using a convenience sampling technique cannot be guaranteed; however, we believe that the results provide us with important data that were not available elsewhere.

## 6. Conclusions

This is the first exploratory study that attempts to identify differences in SARS-CoV-2 viral loads between low- and high-altitude populations. We found that there are no significant differences in terms of viral load between the two populations.

We believe that air quality and its composition, such as the ozone concentration, particulate matter, UV lights, humidity (%), and hypoxia, have no direct relationship with the viral load found in SARS-CoV-2-positive patients; rather, we believe that the differences depend on other factors, such as the stage at which the sample collection was taken and what stage of the viral load curve each patient was in.

Finally, we conclude that men have more viral load than women and that young adults have more viral load than the elderly, and this could be due to a greater and more prolonged exposure to aerosols than those older people or women who surely take more care of themselves and may be more vigilant in the use of biosecurity measures.

## Figures and Tables

**Figure 1 ijerph-19-07945-f001:**
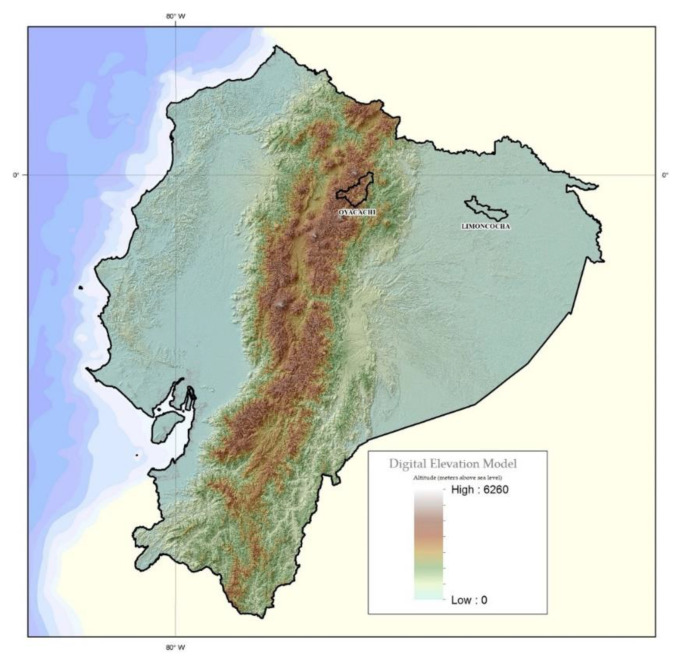
Topographic map of Ecuador highlighting Limoncocha (230 m) and Oyacachi (3800 m).

**Figure 2 ijerph-19-07945-f002:**
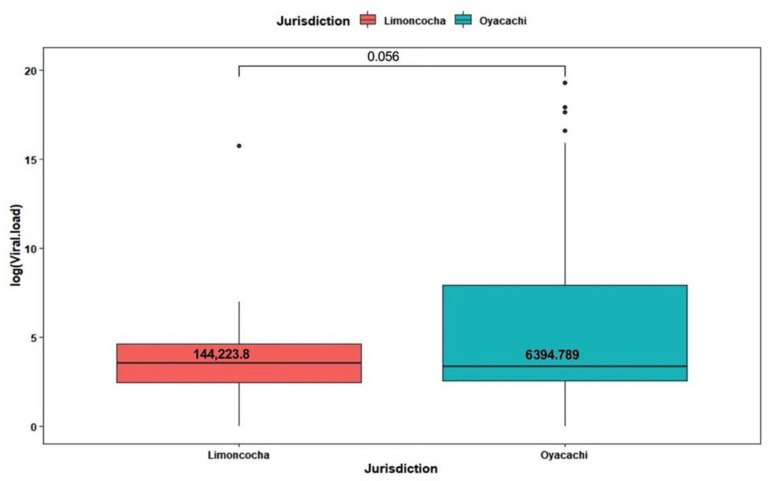
Box plot of viral load composition across jurisdiction (communities) of SARS-CoV-2 test positivity among 108 tested people, 50 from Limoncocha and 58 from Oyacachi.

**Figure 3 ijerph-19-07945-f003:**
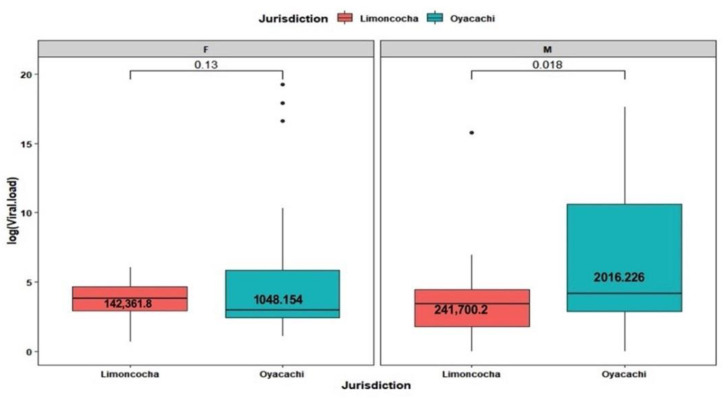
Box plot of viral load composition across jurisdiction (communities) and gender of SARS-CoV-2 test positivity among 108 tested people, 21 women and 29 men from Limoncocha and 30 women and 28 men from Oyacachi.

**Figure 4 ijerph-19-07945-f004:**
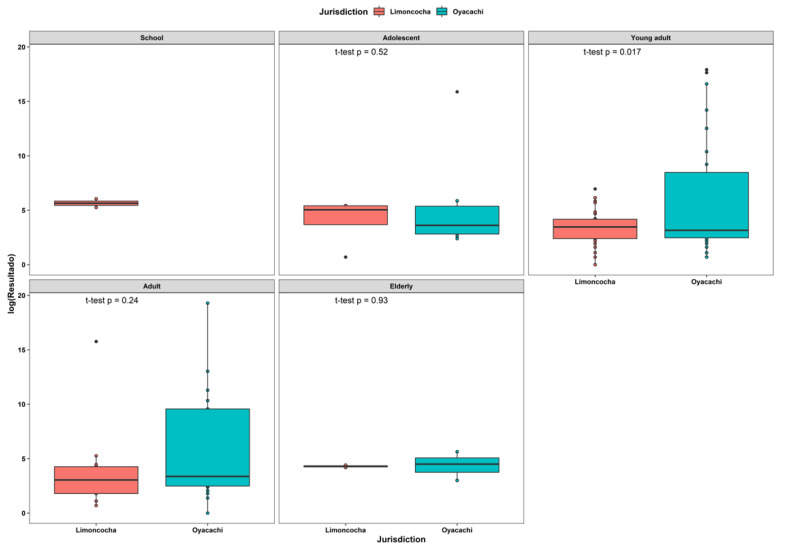
Box plot of viral load composition across age groups among 108 positive tests for SARS-CoV-2 infection. Age groups classification is according to the United Nations guidelines as follows: school (<12 years), adolescent (13–20 years), young adult (21 to 39 years), adult (40 to 64 years), and elderly (>65 years).

**Table 1 ijerph-19-07945-t001:** Viral loads composition across age groups among 108 positive tests for SARS-CoV-2 infection.

	Limoncocha			Oyacachi			
Age Range	N	Viral Load	SD (±)	N	Viral Load	SD (±)	*p*-Value *
**5 to 9**	1	191.0	N/A	0	0	0	N/A
**10 to 14**	1	422.0	N/A	2	3,947,564.0	5,582,629.0	N/A
**15 to 19**	3	110.6	1,105,456.0	2	190.5	229.8	0.64
**20 to 24**	6	57.3	86.7	5	757.6	1667.2	0.64
**25 to 29**	4	124.5	229.7	9	7,093,211.0	15,456,697.0	**0.023**
**30 to 34**	13	149.2	284.6	10	6,049,341.0	19,125,937.0	0.39
**35 to 39**	7	61.4	106.5	6	194.6	430.2	0.51
**40 to 44**	7	1,000,949.0	2,648,156.0	8	125,185.6	204,549.2	0.29
**45 to 49**	4	49	34.6	5	47,526,374.0	1,067,567	0.53
**50 to 54**	0	0	0	3	29.3	14.0	N/A
**55 to 59**	1	44		3	101.6	165.7	N/A
**60 to 64**	1	2		2	15,421.0	21,787.3	N/A
**65 to 69**	1	66		2	185.5	135.0	N/A
**70 to 74**	0	0	0	1	20	N/A	N/A
**>80**	1	82		0	0	0	N/A

An independent *t*-test was used to asses’ differences between groups. We used a 95% confidence level (α = 0.05). * significant values presented in bold

**Table 2 ijerph-19-07945-t002:** Comparative table of main climatological and meteorological differences between Oyacachi and Limoncocha [17,31,32,33,34].

Variable	Oyacachi	Limoncocha
Adult population	739	6817
Road access	Cobblestone road	Asphalt road
Altitude (meters)	3800 to 4300 m	228–2800 m
Barometric pressure (BP)	487 mmHg (65 kPa)	739 mmHg (98 kPa)
Oxygen availability compared to sea level	64%	97%
Weather	Upper montane rainforest	Lowland rainforest
Temperature (°C)	−2–17 °C	18–26 °C
Rainfall (mm)	1200–3000	3200–3400
Relative humidity (%)	89%	>90%
Ozone (O^3^)	27 µg/m^3^	27 µg/m^3^
Particulate matter _2.5_	4 µg/m^3^	5 µg/m^3^
Particulate matter _10_	8 µg/m^3^	8 µg/m^3^
NO_2_	3 µg/m^3^	4 µg/m^3^

## Data Availability

All data is anonymized and available in the following open access repository https://github.com/covid19ec/Altitude (accessed on 30 April 2022).

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
