# Peer review of "SARS-CoV-2 Viral Load Analysis at Low and High Altitude: A Case Study from Ecuador"

_ijerph, 2022, doi:10.3390/ijerph19137945_

Round 1

Reviewer 1 Report

The article presents no differences between the viral loads of people living at low and high altitude in Ecuador and could be published as a case study of Int. J. Environ. Res. Public Health, however, after a serious approach of the authors to the preparation of the article.

The transmission, infectivity and virulence of the virus at high altitudes should theoretically be lower due to environmental factors unfavorable for the virus, but so far it has not been tested experimentally, so any work in this field is noteworthy.

The degree of viremia is the result of many factors, properties and condition of the organism and the environment. The work would be more interesting and complete if the authors obtained environmental data on the differences in ozone concentration, humidity and UV radiation intensity between the studied regions and, above all, include demographic data on population density / frequency of interpersonal contacts. The difference in terms of atmospheric pressure or oxygen concentration for the virus will not be significant. It would be useful to list the environmental differences so that it can be stated which and to what extent are not significant for the viability of the virus in the environment with reference to laboratory experiments.

Comments:

Abstract: the abstract is badly written, very chaotic.

The information presented in the abstract is disordered and requires supplementing and arranging, e.g .:

- "The total number of SARS-CoV-2 positive cases was 108 (40.3%).

"- the number and percentage of positive results were given, but the number of test persons was not given.

- the wording "The attack rate" is not correct and the percentages are given without the number.

- "The mean viral load of the 40.3% (n = 108) 19 patients who tested positive was 3'499,184 copies / ml (SD = 23'931,479 copies / ml)." -

 the data applies to all positive results and the aim of the work is to compare two groups for which no results were presented, which is the most important.

- the sentence "We found no statistically significant differences between groups (p-value: 0.056)" should be summarized after information on test groups and not after information on the mean viral load.

  1. Materials and Methods: the authors used the template for the article and at all costs wanted to complete all the proposed modules, which in their case do not work. This resulted in the duplication of the same information, e.g. :

- point 2.1. Study design does not contain information about the structure and plan of the study.

- point 2.2. Settings and 2.3. Participants contain - exactly the same information.

- points 2.1-2.5 contain the same data in a different form

- 2.6. Statistical methods - contains a description of the diagnostic test. Did anyone read this before posting? The test description does not mention how the threshold was set

- point 2.8. Ethics statement - not necessarily, but can be simplified. All you need is information about the diagnostic screening test and informed written consent of patients to use their results from the study and fulfillment of all legal requirements

- there is no information on the competences of the persons collecting the material and performing the test

- lack of information on the size of the surveyed populations, density of residence, frequency of contacts and economic status

It is necessary to improve the Materials and Methods module so that in a clear, detailed and orderly manner, without duplicating it, it contains all the relevant data: the exact characteristics of the study group, research methods and the manner of performing the study

  1. Results:

 - 3.1. General Results. These are not the results, but a re-duplication of the characteristics of the study population

- 3.2. Age and Sex differences. These are not the results, but a re-duplication of the characteristics of the study population. In terms of age and sex, no results were presented. Where are the age and gender results for infection rates?

- 3.3. Positive testing rates - please provide complete results in an orderly and understandable way for everyone

- line 148-152 - why in italics? - the text is not understandable, it is not known what it is being compared with. Please provide complete data that are compared with each other with the specification of the compared groups

- Fig 2 and Fig 3 - it would be nice to give the mean values ​​in the graphs

- Tab 1- statistical test was used, please specify exactly what kind of test under the table. It is also not known what level was adopted at work, because in 2.6. Statistical methods stated: "We consider the confidence level at α = 0.01, α = 0.05 and α = 0.10." Please consult the statistician which level and where he applied

- Fig 4 - For the age categories shown, the range in years is nowhere given. Please complete in the characteristics of the population and in the diagram

  1. Conclusions :

- The sentence: "Viral load appears to be higher among young adults than among children or the elderly, possibly because they are the most affected population group" is incomprehensible.Is it viremia or the incidence of infections?They are two different things.

The most socially active groups are more exposed to contact, so the infection rate will also be higher.

However, this does not affect the degree of viral load.  Please separate and consider at work these two issues: the degree of viremia - environmentally weakened virus / reduced virulence or generally stronger and more efficient organism by adapting to the demanding environment, and the frequency of infections - depending on factors influencing transmission

Author Response

Point by Point Letter

Submission ID: ijerph-1662446

Dear Editor and reviewers, thank you very much for your effort in observing our manuscript and offering us some comments intended to improve our manuscript.

We have completed a full revision which includes answers to all your comments and suggestions. All changes are highlighted in red within the main manuscript and this point-by-point letter.

Reviewer 1

The article presents no differences between the viral loads of people living at low and high altitude in Ecuador and could be published as a case study of Int. J. Environ. Res. Public Health, however, after a serious approach of the authors to the preparation of the article.

Thanks for your valuable comments.

The transmission, infectivity and virulence of the virus at high altitudes should theoretically be lower due to environmental factors unfavorable for the virus, but so far it has not been tested experimentally, so any work in this field is noteworthy.

Thanks for recognize that, we appreciated your comment, so far, no information is available in terms of viral loads at different altitudes.

The degree of viremia is the result of many factors, properties and condition of the organism and the environment. The work would be more interesting and complete if the authors obtained environmental data on the differences in ozone concentration, humidity and UV radiation intensity between the studied regions and, above all, include demographic data on population density / frequency of interpersonal contacts. The difference in terms of atmospheric pressure or oxygen concentration for the virus will not be significant. It would be useful to list the environmental differences so that it can be stated which and to what extent are not significant for the viability of the virus in the environment with reference to laboratory experiments.

Thanks for your observation. We have added an entire table including the most relevant environmental differences (please refer to table 2). all the changes are highlighted in red within the text.

Comments:

Abstract: the abstract is badly written, very chaotic.

The information presented in the abstract is disordered and requires supplementing and arranging, e.g .:

- "The total number of SARS-CoV-2 positive cases was 108 (40.3%).

"- the number and percentage of positive results were given, but the number of test persons was not given.

- the wording "The attack rate" is not correct and the percentages are given without the number.

- "The mean viral load of the 40.3% (n = 108) 19 patients who tested positive was 3'499,184 copies / ml (SD = 23'931,479 copies / ml)." -

 the data applies to all positive results and the aim of the work is to compare two groups for which no results were presented, which is the most important.

- the sentence "We found no statistically significant differences between groups (p-value: 0.056)" should be summarized after information on test groups and not after information on the mean viral load.

Thanks for pointing this out, we rearranged our wording within the entire abstract and all your suggestions were included.

  1. Materials and Methods: the authors used the template for the article and at all costs wanted to complete all the proposed modules, which in their case do not work. This resulted in the duplication of the same information, e.g. :

- point 2.1. Study design does not contain information about the structure and plan of the study.

- point 2.2. Settings and 2.3. Participants contain - exactly the same information.

- points 2.1-2.5 contain the same data in a different form

- 2.6. Statistical methods - contains a description of the diagnostic test. Did anyone read this before posting? The test description does not mention how the threshold was set

- point 2.8. Ethics statement - not necessarily, but can be simplified. All you need is information about the diagnostic screening test and informed written consent of patients to use their results from the study and fulfillment of all legal requirements

- there is no information on the competences of the persons collecting the material and performing the test

- lack of information on the size of the surveyed populations, density of residence, frequency of contacts and economic status

It is necessary to improve the Materials and Methods module so that in a clear, detailed and orderly manner, without duplicating it, it contains all the relevant data: the exact characteristics of the study group, research methods and the manner of performing the study

The entire methods section was redone, and your observations were all included.

  1. Results:

 - 3.1. General Results. These are not the results, but a re-duplication of the characteristics of the study population.

Thanks, we have clarified this

- 3.2. Age and Sex differences. These are not the results, but a re-duplication of the characteristics of the study population. In terms of age and sex, no results were presented. Where are the age and gender results for infection rates?

An entire 3.5 section entitled Viral Load Analysis by age and sex was incldued

- 3.3. Positive testing rates - please provide complete results in an orderly and understandable way for everyone.

Please refer to table 1

- line 148-152 - why in italics? - the text is not understandable, it is not known what it is being compared with. Please provide complete data that are compared with each other with the specification of the compared groups

That was a mistake, is not in italic anymore

- Fig 2 and Fig 3 - it would be nice to give the mean values ​​in the graphs

We have done it. The figures  now have mean values

- Tab 1- statistical test was used, please specify exactly what kind of test under the table. It is also not known what level was adopted at work, because in 2.6. Statistical methods stated: "We consider the confidence level at α = 0.01, α = 0.05 and α = 0.10." Please consult the statistician which level and where he applied

Thanks for pointing this out, the change was made

- Fig 4 - For the age categories shown, the range in years is nowhere given. Please complete in the characteristics of the population and in the diagram

We have described and specified within the figure legend as follow: Age groups classification follows the United Nations guidelines as follow: School (< 12 years), adolescent (13-20 years), young adult (21 to 39 years), adult (40 to 64 years) and elderly (> 65 years).

  1. Conclusions :

- The sentence: "Viral load appears to be higher among young adults than among children or the elderly, possibly because they are the most affected population group" is incomprehensible.Is it viremia or the incidence of infections?They are two different things.

The most socially active groups are more exposed to contact, so the infection rate will also be higher.

However, this does not affect the degree of viral load.  Please separate and consider at work these two issues: the degree of viremia - environmentally weakened virus / reduced virulence or generally stronger and more efficient organism by adapting to the demanding environment, and the frequency of infections - depending on factors influencing transmission.

The entire conclusion section was redone. The current version is more clear and included your suggestions

Reviewer 2 Report

The study analyzed viral load from COVID-19 patients at low and high altitude in Ecuador. However, there are multiple factors affecting viral load, including age, timing of the samples, disease severity, etc which have not been controlled for. The representativeness and recruitment of the samples were also unclear and the current data and analysis is insufficient to test the main hypothesis.

Comments to the authors:

  1. Study design. The recruitment of participants and testing procedures were not described but could have large impact on the results. The potential difference in sampling time or case finding at Limoncocha and Oyacachi could have explained difference in viral load.
  2. Days of sampling relative to infection or symptom onset has a large impact on viral load. However, this is not analyzed and information may not have been collected.

Author Response

Point by Point Letter

Submission ID: ijerph-1662446

Dear Editor and reviewers, thank you very much for your effort in observing our manuscript and offering us some comments intended to improve our manuscript.

We have completed a full revision which includes answers to all your comments and suggestions. All changes are highlighted in red within the main manuscript and this point-by-point letter.

The study analyzed viral load from COVID-19 patients at low and high altitude in Ecuador. However, there are multiple factors affecting viral load, including age, timing of the samples, disease severity, etc which have not been controlled for. The representativeness and recruitment of the samples were also unclear and the current data and analysis is insufficient to test the main hypothesis.

Thanks for pointing this out, we have expanded our limitation section and incorporated your observations. Nevertheless, we must clarify that People from Limoncocha and Oyacachi are community dwelling individuals and because we performed a single intervention, we were not able to identify information about diseases length.  These types of interventions are large, random samples from a population that in principle is not even suspicious of having COVID-19, therefore, our results are a good proxy of the community situation at the time of the visit.

Comments to the authors:

  1. Study design. The recruitment of participants and testing procedures were not described but could have large impact on the results. The potential difference in sampling time or case finding at Limoncocha and Oyacachi could have explained difference in viral load.

Although this is one of the main imitations, the fact that we responded to the call of a population during the pandemic provided us with a unique opportunity to collect this type of data, however, due to the urgency of the situation, some data such as date of symptom onset were not provided.

  1. Days of sampling relative to infection or symptom onset has a large impact on viral load. However, this is not analyzed and information may not have been collected.

We agree with you, an entire limitation section was incorporated, and your observations accepted.

Reviewer 3 Report

Alenquer et al. evaluated the impact of an environmental factor, altitude, on SARS-CoV-2 infection.  Specifically, the viral load of SARS-CoV-2 in patients living at low and high altitude in Ecuador was compared. In addition to this, demographic characteristics such as gender and age were also considered.

Some statements need to be clarified and further investigated.

These are my main concerns:

  1. Line 34: Please correct “;” with the “.” after [4].
  2. Lines 66-67: “Limoncocha located at 230 m meters above sea level (low altitude) and Oyacachi, located at 3,800 m meters above sea level (high altitude)”. Please report altitude as 280m/ 3800m or 280 metres/ 38000 metres, not 280m metres
  3. Lines 72-73. “Limoncocha located at 230 m meters above sea level and Oyacachi, located at 3,800 m meters above sea level”. The concept has already been clearly stated in the previous paragraph so it is repetitive. The concepts in paragraphs 2.2 and 2.3 could be grouped together in one paragraph.
  4. Lines 80-84: In line with the previous comment, the information reported in this paragraph could be combined with that in paragraphs 2.2 and 2.3.
  5. Lines: 106-109: “As it is detailed in Figure 2, serial…”. Figure 2 does not show the concept described here. Is figure mentioned correct? If not, please provide a clearer and more explicit figure concerning the serial dilutions of the positive control.
  6. Line 130: Please add a full stop at the end of the sentence.
  7. Figure: In all entered figures, the ample size of each compared group is not indicated. Please add this information.
  8. Lines 146-147: “We found no statistically significant differences between communities (p= 0.056) (Figure 2).” The sentence shows a p-value = 0.056 whereas in the figure the p-value is 0.0056. Please enter the correct value. Although the p-value is not significant (>0.05), it is possible to define the 0.056 value as a tendency towards significance. It would be useful to comment more on this difference and, to understand this better, the viral load values could be reported as median (IQR).
  9. Lines 148-152: There are some basic misunderstandings in the reported sentence. Specifically, there is confusion between viral load expressed as copies/ml and ct (threshold cycle). These are two different concepts and therefore the unit of measurement is also different. Please revise the concept and rephrase the sentences.
  10. Lines 157-160: What criteria were used to select the age groups shown in Table 1? The broad stratification chosen meant that many age groups had a very low sample size, which could create a bias in the viral load comparison analysis.
  11. Figure 4: Please report the specific ages of the different groups in addition to the sample sizes.

Author Response

Point by Point Letter

Submission ID: ijerph-1662446

Dear Editor and reviewers, thank you very much for your effort in observing our manuscript and offering us some comments intended to improve our manuscript.

We have completed a full revision which includes answers to all your comments and suggestions. All changes are highlighted in red within the main manuscript and this point-by-point letter.

Alenquer et al. evaluated the impact of an environmental factor, altitude, on SARS-CoV-2 infection.  Specifically, the viral load of SARS-CoV-2 in patients living at low and high altitude in Ecuador was compared. In addition to this, demographic characteristics such as gender and age were also considered.

Thanks for your observations, we have reorganized the entire manuscript for clarity and readability.

Some statements need to be clarified and further investigated.

These are my main concerns:

  1. Line 34: Please correct “;” with the “.” after [4].

Done

  1. Lines 66-67: “Limoncocha located at 230 m meters above sea level (low altitude) and Oyacachi, located at 3,800 m meters above sea level (high altitude)”. Please report altitude as 280m/ 3800m or 280 metres/ 38000 metres, not 280m metres

Thanks, we have done so, thanks for pointing this out

  1. Lines 72-73. “Limoncocha located at 230 m meters above sea level and Oyacachi, located at 3,800 m meters above sea level”. The concept has already been clearly stated in the previous paragraph so it is repetitive. The concepts in paragraphs 2.2 and 2.3 could be grouped together in one paragraph.

Thanks, we accepted your suggestion

  1. Lines 80-84: In line with the previous comment, the information reported in this paragraph could be combined with that in paragraphs 2.2 and 2.3.

Thanks, we accepted your suggestion

  1. Lines: 106-109: “As it is detailed in Figure 2, serial…”. Figure 2 does not show the concept described here. Is figure mentioned correct? If not, please provide a clearer and more explicit figure concerning the serial dilutions of the positive control.

Thanks for your observation, that was a typo. We shortened the paragraph according to one of the other reviewers.

  1. Line 130: Please add a full stop at the end of the sentence.

done

  1. Figure: In all entered figures, the sample size of each compared group is not indicated. Please add this information.

We have done so within the description of each figure

  1. Lines 146-147: “We found no statistically significant differences between communities (p= 0.056) (Figure 2).” The sentence shows a p-value = 0.056 whereas in the figure the p-value is 0.0056. Please enter the correct value. Although the p-value is not significant (>0.05), it is possible to define the 0.056 value as a tendency towards significance. It would be useful to comment more on this difference and, to understand this better, the viral load values could be reported as median (IQR).

Thanks for pointing this out, we have updated the figure

Lines 148-152: There are some basic misunderstandings in the reported sentence. Specifically, there is confusion between viral load expressed as copies/ml and ct (threshold cycle). These are two different concepts and therefore the unit of measurement is also different. Please revise the concept and rephrase the sentences.

Thanks for pointing out. This was a typo mistake. We use the term Ct when we were actually describing viral loads. We only use viral loads as a variable in our manuscript. Viral loads calculation from RT-Ct values are detailed in the methods section.

The new paragraph included is the manuscript is this: "Among female patients, no difference in viral load were found (Viral load mean = 6,165,595 copies/ml [p-value of 0.13]). When compared low versus high altitude viral load values, men from Oyacachi had a higher viral load (mean 2,016,226 copies/ml) than those men living in Limoncocha (mean 241,700 copies/ml), being this difference statistically significant at confidence level of α=0.05 (p-value of 0.018) (figure 3).

  1. Lines 157-160: What criteria were used to select the age groups shown in Table 1? The broad stratification chosen meant that many age groups had a very low sample size, which could create a bias in the viral load comparison analysis.

We have clarified the categorization

  1. Figure 4: Please report the specific ages of the different groups in addition to the sample sizes.

We have done so, thanks for your observation